# Biotechnology of the Tree Fern *Cyathea smithii* (J.D. Hooker; Soft Tree Fern, Katote) II Cell Suspension Culture: Focusing on Structure and Physiology in the Presence of 2,4-D and BAP

**DOI:** 10.3390/cells11091396

**Published:** 2022-04-20

**Authors:** Jan J. Rybczyński, Andrzej Kaźmierczak, Katarzyna Dos Santos Szewczyk, Wojciech Tomaszewicz, Małgorzata Miazga-Karska, Anna Mikuła

**Affiliations:** 1Polish Academy of Sciences, Botanical Garden-Center for Biology Diversity Conservation in Powsin, 2 Prawdziwka Str., 02-973 Warsaw, Poland; w.tomaszewicz@obpan.pl (W.T.); a.mikula@obpan.pl (A.M.); 2Department of Cytophysiology, Łódź University, 90-236 Lódź, Poland; andrzej.kazmierczak@biol.uni.lodz.pl; 3Department of Pharmaceutical Botany, Faculty of Pharmacy, Medical University of Lublin, 1 Chodźki Str., 20-093 Lublin, Poland; k.szewczyk@umlub.pl; 4Department of Biochemistry and Biotechnology, Medical University of Lublin, 20-093 Lublin, Poland; malgorzata.miazga-karska@umlub.pl

**Keywords:** antibacterial activity of tissue isolates, ethylene production, nuclear DNA content, plant cell, primary and secondary metabolites, structural analysis, sugars effect

## Abstract

The aim of our research was to describe the structure and growth potential of a cell suspension of the tree fern *Cyathea smithii*. Experiments were performed on an established cell suspension with ½ MS medium supplemented with 9.05 µM 2,4-D + 0.88 µM BAP. In the experiments, attention was paid to the microscopic description of cell suspension, evaluation of cell growth dependent on the initial mass of cells and organic carbon source in the medium, the length of the passage, the content of one selected flavonoid in the post-culture medium, nuclear DNA content, ethylene production, and the antimicrobial value of the extract. For a better understanding of the cell changes that occurred during the culture of the suspension, the following structures of the cell were observed: nucleus, lipid bodies, tannin deposits, starch grains, cell walls, primary lamina, and the filaments of metabolites released into the medium. The nuclear DNA content (acriflavine-Feulgen staining) of cell aggregates distinctly indicated a lack of changes in the sporophytic origin of the cultured cell suspension. The physiological activity of the suspension was found to be high because of kinetics, intensive production of ethylene, and quercetin production. The microbiological studies suggested that the cell suspension possessed a bactericidal character against microaerobic Gram-positive bacteria. A sample of the cell suspension showed bacteriostatic activity against aerobic bacteria.

## 1. Introduction

In the world of ferns, many species reproduce vegetatively in nature and in vitro. Tree ferns are characterized by the formation of a trunk and do not form vegetative organs like stolons, gemmae, tubers, or bulbils, but aerial layering sometimes sticks to their trunks, which helps to form roots [1]. Offsets are the only method of vegetative propagation by tree ferns through new organism formation on the surface of the trunk and have been explored by horticulturists [2]. The number of the papers on plant cell manipulations is limited to only a few species of ferns. This paper will present our attempt to establish new fields of experimental plant biology for a selected tree fern species, namely, *Cyathea smithii* Hook.f.

The genus *Cyathea* comprises over 470 species, which have seldom been used for in vitro culture and biotechnology experiments. The list includes the following species: *Cyathea dregei* [3], *C. australis* [4], *C. capensis*, *C. cooperi*, *C. brownie*, *C. dealbata*, *C. dregei*, *C. leichhardtiana*, *C. robertsiana*, *C. schanschin*, *C. smithii* [5], *C. atrovirens* [6,7], *C. spinulosa* [8,9], *C. gigantea* [10], and *C. delgadii* [11]. The last one of these recently became an experimental model for exploring somatic embryogenesis [12,13,14]. The vegetative propagation by spore germination and mass production of gametophyte clamps on agar medium has been described as a tool for the effective sporophyte production of 11 species of *Cyathea* [5]. The development of cell suspension cultures is an alternative route for plant regeneration. It has been successful in numerous seed plants, but there are only few reports for ferns, and none concern tree ferns. The sporophyte- or gametophyte-derived tissue callus was the plant material used for the establishment of these cell suspensions: *Nephrolepis exaltata* [15,16] and *Platycerium coronarium* [17]. In the cases of *Platycerium bifurcatum* [18,19] and *Dryopteris affinis* [20], specially designed experiments (based on the mechanical maceration of the leaf blade and different sizes of gametophytes, respectively) were conducted for the initiation of liquid cultures but not “classical” cell suspension.

Ferns are one of several plant groups whose products of metabolism are of interest as a source of human and animal food [4,5]. The best example is *Cyathea australis*, whose trunk is used as source of carbohydrates by Aboriginal Australians because of the very rich deposits of starch within its parenchyma cells. Plants produce a wide range of low molecular weight, natural products via a network of typically complex metabolic pathways. These substances include dyes, pigments, flavors, aromas, medicines, and poisons [21], but the following groups of chemical metabolites have also been listed in references relating to ferns: saccharides, polyketides, terpenoids, nitrogen-containing metabolites, and phenolics. Qualitative and quantitative analyses of extracts of the various organs of green plants of numerous *Cyathea* species have been published over the last 70 years [22,23]. Special attention has been paid to chemical constituents with underlying flavonoids for *C. fauriei* and *C. hancockii* [22]; *C. fauriei* [24]; *C. fauriei*, *C. mertensiana*, *C. leichhardtiana*, *C. podophylla*, and *C. hancockii* [25]; *C. contaminans* [26]; and *C. nilgirensis*, *C. gigantea* and *C. crinite* [27,28]. Very intense yellow coloration of the tissue culture and medium following the growth of *Cyathea* cell suspensions indicated that our attention should be paid to flavonoids (polyphenols), which were observed in our liquid culture but not on solid agar media. Our primary analysis (data not published) pointed to quercetin characterized by the ultraviolet absorption spectrum of 301 nm [29] as the flavonoid to evaluate the physiological activity of the *C. smithii* cell suspension.

In cell suspensions, the plant material is totally submerged in a liquid medium, and it depends on moderate aeration of the latter for its respiratory needs. Under these conditions plant material seems to be independent of the plant growth hormone conditions of the initial tissue but dependent on the experimentally determined conditions of the culture. Agitation of the suspension culture has some technical advantages such as improved uniformity of cell growth, easier control of the culture process, and easier cultivation on a larger scale. By replacing an equal volume of old medium with fresh medium, the culture medium composition is kept constant. This type of culture began in the 1950s and was performed on various plant species, initially marigold, tobacco [30,31], sunflower, grape [32], bean [33], pea [34,35], and carrot [36]. However, the great amount of information now available on cell suspension cultures made it extremely difficult to find the original data dealing with this topic, although it appears that Muir and co-authors [30] were the first to employ this method. Development of this method of plant culture has been very promising, and it has finally established the technical background for work on the direct transformation of the genome of single cells, protoplasts [37], and microspores [38], which can never be accomplished on solid media. Studies on the liquid culture of plant cells, especially those on cellular metabolism, and more recently, biopharmaceutical production, have improved our understanding of plant cell physiology and provided a new source of secondary metabolites and proteins [39,40]. This enormous increase in the discovery of new secondary metabolic pathways and the production of these compounds has mainly occurred in dicots, rarely monocots, and in very few cryptogamic plant species. The main reason for this is that the morphogenic potential of the fern body under in vitro conditions is poorly understood [41]. Furthermore, few papers have been published on the establishment of cell suspensions initiated from proliferated explants producing callus tissue that originated from growing sporophytes or gametophytes in axenic culture. With the exception of sporophyte explants, which were rarely used, these included *N. exaltata* [15,16], *P. coronarium* [17], and *P. bifurcatum* [18,19], and from a single cell to few-cell complexes of gametophytes of *Dryopteris affinis* [20].

Among the different types of plant growth hormones, the only one in gaseous form is ethylene (C_2_H_2_; ETH), which is produced by dividing cells. It is regarded as a multifunctional, biologically active organic controlling many physiological and developmental processes in plants. The in vitro method provided favorable conditions for two types of studies on ETH’s effect on culture: an analysis of ethylene intensity production and the controlling and alternatively direct effect of ETH on the metabolism of cells [42,43] or organs [44]. Intensive studies carried out on these effects produced a list of organic and nonorganic substances that are used to control ETH production in the living culture of plant cells, organs, regenerants, and whole plantlets. The initial data on ETH production evaluation in plant cell suspension showed a connection between complexes of the following species: soybean, wheat, sweet clover, rue, and *Haplopappus gracilis*. During this time, variation in production during the growing cycle and variation among the analyzed species were studied [45]. ETH may function in many types of plant [46,47] tissue culture systems as either a promoter or inhibitor of physiological processes, depending on the species studied [48]. At present, attention is also being paid to control the production of ETH in different systems of cyanobacterial artificial biofilms [49]. However, in the case of fern biotechnology, we should take notice of studies on the effect of ETH on spore germination in light and darkness [50]. Spores of *Onoclea sensibilis* were most sensitive to ETH; however, ethylene did not directly inhibit DNA replication during the stages of germination prior to DNA synthesis but blocked germination at an earlier fundamental step [51]. The inhibitory effect of ethylene on plant regeneration from frond and rhizome pieces of *Platycerium coronarium* cultured on solidified Gelrite medium was shown. This indicated that the biosynthetic pathway of ETH appeared to be different from that of other plants but similar to that of some other ferns [52]. Enhancing the role of ETH in apospory and regeneration from young sporophytic leaves was undertaken in experiments on *Ceratopteris richardii* [53].

Due to the lack of studies on the cell suspension of ferns, the aim of this paper is to show the complex results of experiments carried out on a cell suspension of tree fern *C. smithii*, concentrating on the structural and physiological levels, nuclear DNA content, and antibacterial value of the organic isolates of the cell aggregates.

## 2. Materials and Methods

### 2.1. Medium Description

In all experiments, plant material was cultured on ½ MS mineral medium [54]. Plant growth hormone supplementation was changed according to the needs of the experiments undertaken. Physical values of the medium ranged as follows: before autoclave sterilization, pH = 5.8, and after autoclave treatment, pH = 5.32; its conductivity was 97.0 mV (H_2_O double distilled: pH = 5.73, conductivity 74.4 mV).

### 2.2. Culture Initiation and Cell Suspension Establishment

Tree fern *C. smithii* was propagated from young sporophytes regenerated via spontaneous syngamy in long-term gametophyte culture. The gametophyte cultures were maintained on ½ (half-strength) MS medium [54] supplemented with 2% sucrose and adjusted to pH 5.8 prior to autoclaving. Two- to three-leaved, very small sporophytes were isolated from the surrounding gametophytes and transferred into new jars containing ½ MS medium to allow for further growth. Plants measuring about 4 to 5 cm long with well-developed roots were the source of explants. For callus induction, explants were cultured on MS medium supplemented with 2,4-D (dichlorophenoxyacetic acid: 0.5, 1.0, and 2.0 mg/L) and BAP (benzylaminopurine) (0.2 and 2.0 mg/L) in six combinations. Cultures were maintained in deep Petri dishes containing about 50 mL of medium and kept in a phytotron in the dark. For cell suspension establishment, the callus was transferred to liquid MS medium. The development of suspension was followed by increasing the volume of the medium by 10, 20, 40, and 80 mL increments in a 250 mL conical flask at two-week-long intervals. Flasks were plugged with cotton wool and covered with a protective paper cap. The suspension culture was produced by shaking at 120 rpm on a rotary horizontal shaker (Infors Rt 250, Switzerland) of 3.0 cm amplitude. Cell suspensions were sub-cultured at two-week intervals using fresh medium at a temper replacement of an equal volume of the medium. All cultures were subjected to a photoperiod of 16/8 h day/night with diffused light. The culture was maintained under a constant temperature of 22 ± 10 °C in a phytotron. All in vitro manipulations were done under sterile conditions beneath a flow-hood (Polon, Poland). For any cyto-morphological studies, suspensions require washing twice with sterile medium/water to remove culture debris and medium and substances that could be secreted into the medium.

### 2.3. Preparation of Living and Fixed Specimens for Bright-Field Light Microscopy

Samples of various ages of culture were isolated from cell suspensions, and living specimens were observed using Olympus AHBT-3 Vanox and Vert JMT2 (Olympus, Tokyo, Japan) bright-field light microscopes, both equipped with Nomarski contrast. For starch grain, lipid, and nucleus analyses, the sample of tissue was washed out from the medium and later fixed with FAA (formaldehyde + glacial acetic acid + ethanol) (5:5:90, respectively). Subsequently, a sample of the suspension was transferred onto a slide and stained with IKI (iodine in aqueous potassium iodide solution) or Sudan III and IV, or 2% acetocarmine (a saturated solution of carmine in 45% acetic acid); finally, a coverslip was applied.

### 2.4. Scanning Electron Microscopy (SEM) Preparation

For a better understanding of the cytological and structural changes that occur during the liquid culture of cells and aggregate culture, SEM was employed. The procedure consisted of fixing samples isolated from a two-week-old subculture in 3% (*v*/*v*) glutaraldehyde in 0.1 mol/L cacodylate buffer at pH 7.0 for 2 h followed by 2.0% (*w*/*v*) osmium tetroxide (OsO_4_) for 24 h in a refrigerator. The samples were then dehydrated using a graded series of ethanol (from 10 to 99.6% (*v*/*v*)) and acetone and dried using liquid CO_2_ in a Type E 3100 Jambo Series II critical-point dryer (Polarno Equipment LTD, Lincoln, England). Samples were then coated with gold in a coater (Jeol, Tokyo, Japan). Finally, samples were scanned using a Jeol model JSM-S1 scanning electron microscope (JEOL Ltd., Tokyo, Japan) [55].

### 2.5. Semi-Thin Specimen Preparation

Two-week-old cell suspensions were sampled and fixed in 2% (*w*/*v*) paraformaldehyde (Fluka, Buchs, Swiss) and 3% (*v*/*v*) glutaraldehyde (Sigma, St. Louis, MO, USA) in 0.05 M sodium cacodylate buffer (Fluka) at pH 7.2 at room temperature for 4 h. The samples were rinsed three times in the same buffer, and post-fixed in 2% (*w*/*v*) OsO_4_ (Carl Roth, Karlsruhe, Germany) in 0.05 M cacodylate buffer at 4 °C for 6 h. After rinsing in 0.05 M cacodylate buffer, explants were dehydrated in a graded ethanol series (in 10% increments), followed by absolute ethanol and propylene oxide. The samples were infiltrated in graded EPON epoxy resin (Sigma) mixtures for 48 h in total. The samples were transferred to flat embedding molds, and the resin was polymerized at 65 °C for 16 h. Semi-thin (2 µm) sections were cut using a Leica Ultracut E ultramicrotome (Leica, Wetzlar, Germany). Semi-thin sections were stained with aqueous 0.1% (*w*/*v*) toluidine blue in 1% (*w*/*v*) sodium tetraborate solution and were analyzed by means of a light microscope (see above).

### 2.6. Preparation of Living Specimens for Confocal Microscopy

Living specimens were prepared from the two-week-old cell suspension to identify the distribution of flavonoids and tannins in the cells. The observations were made with the help of a Zeiss Axiovert 200 M microscope combined with the Zeiss LSM510 confocal system. Before cell suspension aggregate location on the surface of the glass slides, the plant material was washed three times in distilled water to remove the culture medium enriched by the suspension of flavonoids and other substances in it.

### 2.7. Acriflavine Nuclei Staining of Cell Suspension and Determination of Nuclear DNA Content

Water-washed, medium culture-free, and unfixed intact cells of suspension were hydrolyzed in 4 N HCl for 20–30 min at 30 °C, rinsed once in water, and stained in a solution containing acriflavine (Sigma, 200 μg/mL) and K_2_S_2_O_5_ (5 mg/mL in 0.1 N HCl) for 30 min at 30 °C. The stained cells were washed three times (3–5 min each) in a concentrated HCl–70% methanol mixture (2:98 *v*/*v* at 30 °C), which removed the non-covalently bound stain from cells, and twice in distilled water. Cells were finally transferred into a drop of 10% glycerol and covered with a coverslip. The stained cells were examined with an epifluorescence microscope (Optiphot-2, Nikon, Japan) equipped with ACT-1 software (Precoptic, Poland) and a B2A filter generating the blue light of the mercury lamp (Osram; [56]). The fluorescence intensity of the nuclei reflecting the DNA content was quantified from the digital images using the ImageJ software (https://imagej.nih.gov, accessed on 7 October 2021), and the channels of the RGB image were split using the “split channel” tool. The red channel was then thresholded such that the nuclei were highlighted. Using the tracing tool, each nucleus was then individually selected to measure its integrated density. Subsequently, frequency distributions over fluorescence intensity were constructed, which allowed us to infer the stage in the cell cycle both in the *C. smithii* fern and in the meristem of *Zea mays* roots used as a standard.

### 2.8. Cell Suspension Maintenance

For the experiments described in this paper, cells originated in an established cell suspension cultured in ½ MS medium supplemented with 9.05 µM 2,4-D + 0.88 µM BAP in a regular two-week-long subculture with 3 g of tissue that was regularly transferred.

### 2.9. Fresh, Dry, and Ash Matter Evaluation

The contents of the conical flask containing the cell aggregates and liquid culture medium were collected by paper filtration, drained, and balanced. To determine the dry matter data, the samples were then dried at 65 °C to acquire constant balance. Mineralization of the dried cell suspension was performed in a muffle furnace (Nabertherm L40/11/P320) using the following time/temperature procedure: 120 °C/2 h, 200 °C/1 h, 300 °C/1 h, and 450 °C/5 h [57]. After that, the samples were balanced.

### 2.10. Measurements of the Growth Kinetics of Cell Suspensions Depending on the Type of Energy Source

The experiments were carried out in a fifteen-day-long culture period, which was maintained in a fourteen-day-long subculture over the last 3 years.

Briefly, cultures were maintained in a 250 mL conical flask with an initial medium value of 80 mL ½ MS [54] supplemented with 9.05 µM 2,4-D + 0.88 µM BAP (all Sigma chemicals). Experiments were repeated in the flask per each culture period with a 3.0 g initial mass of cell suspension. Each time the cultured tissue was derived from the same sample, three consecutive two-week-long subcultures were created with the same medium values (3.0 g of tissue per 80 mL of medium). At each starting point the tissue sample was the same age (14 d old). Four different sugars were considered for the evaluation of the growth kinetics of the cell suspension: sucrose (fructose + glucose), maltose (glucose + glucose), and glucose and fructose individually. Three determinants of cell growth were evaluated: fresh and dry matter and weight of ash. Culture conditions and cell maintenance were carried out as described above.

### 2.11. Measurements of Quercetin Produced by the Cell Suspension during the 15-Day Subculture in the Post-Culture Medium

At the end of the subculture, i.e., after 15 days of culture, the medium was separated from the plant material by filtration with white Whatman paper. The post-culture medium was centrifugated at 10,000 rpm for 20 min with a Janecky centrifuge. To measure the extinction of cell debris, the supernatant was transferred to a cuvette for spectral analysis. A GeneQuant 1300 spectrometer (General Electric, Sweden) was used for the evaluation of extinction at 301 nm. Quercetin [58] was selected for the analysis of flavonoids on the basis of the as-yet unpublished results of an analysis of the organic components of the post-culture medium. Cultures were carried out in medium supplemented with four sugars: sucrose, maltose (disaccharides), and glucose and fructose (monosaccharides).

### 2.12. Effectivity of Ethylene Production in Relation to the Fresh Weight of Initial Tissue

Cultures were maintained in the presence of 40 mL of the medium (see above) in a 250 mL conical flask. The following samples of drained tissue from fourteen-day-old cultures were studied: 0.5, 1.0, 1.5, 2.0, and 3.0 g of fresh weight (FW). After 15 days, cultures were completed, and ethylene, fresh and dry weight, and weight of ash were evaluated. For ethylene evaluation, the 250 mL conical flasks were tightly capped with the inserted pipette tips and connected via flexible tubes to the pump of an SCS56 handheld ethylene analyzer (Storage Control Systems, Sparta, UK). Readings of the ethylene production in ppm were taken.

### 2.13. Effectivity of Ethylene Production According to the Length of Time

To evaluate the cell suspension growth, ethylene production was employed. To this end, in the following cell suspension, cultures after 15, 12, 9, 6, and 3 days were collected in “Zero” time. At each day point the fourteen-day-old cell suspension was used for evaluation of cell effectivity.

### 2.14. Antibacterial Activity of Extracts of Cell Suspension

*C. smithii* extracts were prepared by sonication at a controlled temperature (40 ± 2 °C) for 30 min with a mixture of methanol–acetone–water (3:1:1, *v*/*v*/*v*; 3 × 50 mL) to determine antibacterial activity. Then the combined extracts were filtered, evaporated under reduced pressure, and lyophilized in a vacuum concentrator (FreeZone 1 Liter apparatus; Labconco, Kansas City, MO, USA) to obtain dried residues. All tested samples were screened for their in vitro antibacterial activity against these strains: aerobic Gram-positive *Staphylococcus aureus* ATCC 25923, *Staphylococcus epidermidis* ATCC 12228, aerobic Gram-negative *Escherichia coli* ATCC 25992, *Pseudomonas aeruginosa* ATCC 27853, microaerobic Gram-positive *Propionibacterium acnes* PCM 2400, *Propionibacterium acnes* PCM 2334, *S. mutans* PCM 2502, and *S. sanguinis* PCM 2335. For antibacterial activity determination we used Mueller–Hinton agar or broth (MH-agar, MH-broth) for aerobic and Brain Heart Infusion agar or broth (BHI-agar, BHI-broth) for microaerobic strains. Bacterial inoculum was prepared by subculturing microorganisms into MH-agar or BHI-agar at 37 °C for 24 or 48 h, respectively. The growth was harvested using 5 mL of 0.9% NaCl and diluted to 0.5 McFarland [1.5 × 108 CFU/mL (CFU: colony-forming unit)].

#### 2.14.1. Antibacterial Character Determined via Agar Disc Diffusion Assay

The antibacterial character of all samples was initially determined by a disc diffusion assay [59]. The bacterial inoculum (using a cotton swab) was spread on the agar surface of the Petri plates. The solutions of all tested samples (100 µg/mL in DMSO) were placed on inoculated Petri plates. Plates with MH-agar (for aerobic strains) were incubated for 24 h at 37 °C and plates with BHI-agar for 48 h at 37 °C. The diameter of the growth inhibition zone around the sample was measured after incubation.

#### 2.14.2. MIC (Minimum Inhibitory Concentration) and MBC (Minimal Bactericidal Concentration) Analyses

The minimum inhibitory concentration (MIC) of plant samples was determined for bacterial strains that exhibited bacterial growth inhibition zones. The test was performed using double serial microdilution in the 96-well microtiter plates according to the CLSI method with some modifications [60]. A total of 200 µL of broth was pipetted into each well. The double serial dilution of the tested derivatives was performed in the test wells, causing rises in concentrations ranging from 0.098 to 2000 µg/mL. Finally, 2 µL of tested bacteria inoculum was added to the wells (except the negative sterility control and blank dye control). The tests were performed at 37 °C for 24 h (aerobic strains) or for 48 h (microaerobic strains). After incubation, the panel was digitally analyzed at 600 nm using the Bio Tek Synergy microplate reader (BioTek, Winooski, VT, USA). The growth intensity in the sample well was compared with the controls: negative, dye, and positive. Additionally, the MBC (minimal bactericidal concentration) was determined by spreading on agar medium (10 µL) samples from the clear well that did not show any growth after incubation in the MIC test. The plates were incubated at 37 °C for 24 h, and the MBC was defined as the lowest concentration of sample without bacterial growth. Each experiment was repeated in triplicate.

## 3. Results

### 3.1. Microscopic Analysis of Cell Aggregates

An optimal callus proliferation for root explants occurred with 9.05 µM 2,4-D + 0.88 µM BAP. Figure 1A shows roots in the long-term culture of one of the tree ferns studied here that were the source of the explants. The root explant response is presented in Figure 1B. Callus proliferation was initiated by cell division of the primary cortex, which resulted in callus formation (Figure 1C). Proliferated callus transferred to liquid medium (Figure 1D) initially produced a cell suspension comprising individual cells or few-cell aggregates (Figure 2A). The two-week interval between subcultures was very crucial for the long-term maintenance of the cell suspensions. Dry weight and ash represented about one-tenth and one-hundredth of the FW, respectively.

Microscopic analysis of cell aggregates revealed a rich accumulation of starch and distinct nuclei with nucleoli (Figure 2). The proliferating cells (Figure 2A) contained a distinct nucleus, usually with one small, dense, spherical nucleolus (Figure 2B). Globular lipid bodies were recognized with the help of Sudan III staining (Figure 2C). The cytoplasm contained large complexes of amyloplasts with starch grains in various shapes and sizes (Figure 2D). An extension of the culture over two weeks (without subculture) resulted in brown coloration of the cell aggregates and medium and structure degradation (Figure 2E).

Actively dividing cells were characterized by regular cell divisions, resulting in larger aggregates and an increase in fresh weight (Figure 3A,B). The SEM studies enabled analysis of cell aggregates (Figure 3), providing ample evidence of regular cell divisions that represented a very early stage of embryogenic e in the cell suspension. The structure of small cell aggregates showed evidence of four separated cell clusters (Figure 3C,D). Their further development led to the emergence of compact regular structures (Figure 3E,F). In the sections, the most centrally located cell was triangular with three thin walls enclosed by a “crown” of cells, each having a different shape.

Figure 4 shows the internal structure of cells comprising the cell aggregate. Thin cell walls separated the cells of small cell aggregates by a thin middle lamella. The cell wall lacked any indication of secondary cell wall formation. The cells comprised a large vacuole and cytoplasm with numerous organelles. The organelles were arranged in chains along the cell wall (Figure 4A,B) or narrow bands of the cytoplasm (Figure 4C). Starch grains within the plastids, globular structure, and electron-dense bodies (tannin) occurred in different cells. In some of them, large deposits of starch were visible in different sizes and shapes (Figure 4D).

With the help of auto fluorescence, an excitation of the metabolic activity of the cytoplasm located close to the cell wall was illuminated (Figure 5A,B). Senescent cells provided evidence of the copious production of metabolites located on the outer surface of the cell wall. Figure 5C shows the fibrillar and granular character of the metabolic products (flavones) produced by the cell. Figure 5D confirms intense metabolite production (flavones) in cells that lost their potential to divide. The cell wall surface was surrounded by the numerous filaments. The second part of metabolite production was located inside the cell vacuole, producing electron-dense bodies—tannins (Figure 5E,F). Usage of the culture medium supplemented with a high concentration of the purine type of cytokinin BAP (8.8 µM) resulted in darkening of the culture and necrosis of the cell aggregates, which was observed during the next two lots of subculturing (4 weeks). The necrosis was initiated by cells with a high level of vacuolization among those forming the particular aggregate.

### 3.2. Morphogenic Potential Evaluation of Cell Suspensions

The transfer of the *C. smithii* suspension to regeneration agar-solidified ½ MS medium supplemented with auxin (NAA) and cytokinins such as kinetin, zeatin, and thidiazuron in different concentrations resulted in cell elongation, accumulation of starch grains, and formation of globular structures (tannins). After 4 weeks of culture the implanted tissue turned brown, and no cell division occurred. Regardless of light conditions, no morphogenic processes were observed. An application of cell suspension in alginate capsules helped to carry out numerous tests but did not have an effect on the induction of any morphogenic processes (Figure 6).

### 3.3. Measurement of Cell Suspension Kinetics

In these experiments attention was paid to: (a) the evaluation of cell growth dependent on the initial mass of cells and the organic carbon source in the medium, (b) length of the passage, (c) the content of one selected flavonoid in the post-culture medium according to the initial mass of cells, (d) the DNA content of the nucleus of cells in the suspension, and (e) ethylene production.

Figure 7 presents the kinetic of standard growth of *C. smithii* cell suspension as the main source of the cells used in other programs of experiments. The response of the cultured cell suspension was strongly correlated with the fresh mass of its initiating sample. The highest values were obtained for 3.0 g tissue for all studied saccharides, with fructose the best (Figure 8). When comparing fresh and dry masses and ash, the ratios of these parameters were almost 10 and 100 times smaller, respectively. The studied saccharides sample of 3.0 g tissue appeared optimal for all three studied parameters. Among the studied saccharides, there were significant differences between both disaccharides, especially for sucrose and maltose content. For monosaccharides, fructose stimulated the growth of all studied parameters better than glucose. The most intensive growth of parameters was observed in the cases of sucrose and fructose.

The relation of the time period to the growth of the cell mass parameters was as follows: fresh mass, dry mass, and ash weight ranged 10 > 10 > 10. The highest value of the growth intensity occurred on the twelfth day of the culture.

### 3.4. Nuclear DNA Content of Cell Suspension Determined with the Help of DNA-Specific Fluorochrome Acriflavine with Acetic Hydrolysis

Acriflavine-stained nuclei (Figure 9) showed good contrast and for the established cell suspension derived from gametophyte G1 ranged from 40 to 50 pg, while those for G2 ranged from 100 to 110 pg. For the cell suspension obtained from sporophyte, G1 presented 70 pg, while G2 was evaluated at 160 pg. The above-mentioned data were obtained in the presence of the integral standard for *Zea mays*, G2 = 106–127 pg.

### 3.5. Effectivity of ETH Production in Relation to the Fresh Weight of Initial Tissue of Subculture

The results of the measurements showed that the amount of ETH was greatest on the third day of culture with the lowest FW and DW, while it was lowest on the sixth day of culture. This fact indicated that cells with a lower initial weight produced a higher amount of ETH, but those with a greater initial weight produced a lower amount of ETH. When the calculation was carried out for ash, the relationship was opposite to that in the DW and FW. This occurred because, in a culture period with a lower initial number of cells, the cells actively divided, but in a culture period with a greater initial number of cells, the (probable) stabilization of the cell culture (i.e., plateau) dominated over cell division (Table 1).

### 3.6. Effectivity of ETH Production According to the Length of Time

The results of the measurements showed that the amount of ethylene was low on the twelfth day of culture, with the greatest FW and DW, compared to third, sixth, ninth, and fifteenth days of the culture. This fact indicated that cell cultures younger and older than 12 days produced more ETH. On the twelfth day of the culture the amount of ETH was greatest compared to the other days when the calculation was made per day of culture. In the case of ash, the correlation was opposite to the other two parameters (fresh and dry matter). This was related to the fact that on the third to ninth days of culture the cells actively divided, but on the fifteenth day of culture the senescence process existed (Figure 1) (Table 2).

### 3.7. Antibacterial Activity of C. smithii Extracts Tested with a Modified Disc Diffusion Method

The size of the bacterial growth inhibition zone was directly proportional to the degree of bacterial sensitivity to the tested drug. The bigger the inhibition zone, the more sensitive bacteria were to the analyzed agent. A sample of the cell suspension showed the broadest spectrum of activity against all tested aerobic, anaerobic, Gram-positive, and Gram-negative strains (Table 3). It had the strongest effect on aerobic Gram-negative strains (creating growth inhibition zones in the range of 36 to 40 mm), more than on microaerobic strains (16–22 mm) and aerobic Gram-positive strains (10–15 mm). Most importantly, the cell suspension sample showed more than three times greater activity against Gram-negative strains than the gallic acid used as the standard in this experiment.

Flavonoid aglycones (quercetin, luteolin) and phenolic acid (gallic acid, *p*-coumaric acid, caffeic acid and vanillic acid) standards used in our experiment—except for gallic acid—had no activity against the tested bacterial strains. Gallic acid showed a broad spectrum of activity against all strains in the range of 10 to 36 mm (Table 3).

Moreover, the MIC (minimum inhibitory concentration) of agents was determined for active extracts that exhibited bacterial growth inhibition zones. The MIC values analysis in Table 4 confirmed that the most favorable MIC values (MIC 7.8–125 μg/mL) among the examined samples against tested strains were seen in the cell suspension sample.

In the MIC test, the activity of *C. smithii* extracts was determined in the range of 250–500 against anaerobic strains. The remaining sample determinations against the combination of microorganisms showed no significant activity. Additionally, to distinguish the inhibiting and killing abilities of tested extracts, the MBC (minimal bactericidal concentration) was determined.

The MIC is the lowest concentration of drug-inhibiting bacterial growth that will have no turbidity in culture media. Additionally, the MBC is the lowest concentration that kills bacteria. It is known that bacteriostatic antimicrobial agents have an MBC/MIC ratio greater than or equal to 16 for a given bacterium, while for bactericidal antimicrobial agents the MBC/MIC ratio is less than or equal to 4 [61]. The MBC/MIC ratios for the tested strains in this study (Table 4) suggested that the cell suspension possessed a bactericidal character against microaerobic Gram-positive bacteria. Samples of cell suspension against aerobic bacteria showed bacteriostatic activity. When used as one of the positive controls, gallic acid exhibited a bacteriostatic mode of action against the tested pathogens.

Production of the flavonoid quercetin with the wavelength 301 nm was correlated to the source of C-organics and the initial weight of tissue. In general, the value of the extinction parameter showed a constant increase correlated with the initial weight of tissue and used sugars. In the case of sucrose, the extinction parameter appeared to be optimal for 1.5 to 3.0 g of the initial mass of cultured cells. However, the highest extinction value was obtained for medium supplemented with fructose in the presence of 3.0 g of tissue initiating the culture (Figure 10). The suspension cultured in the presence of maltose showed the lowest level of extinction for all tissue weight parameters studied. In the presence of maltose, the tissue showed a 2.5-fold lower value than the optimal value shown by fructose.

## 4. Discussion

Herein, we provide evidence of the possibility to develop and establish a tree fern cell suspension with the help of various methods. The presented results concern the structural analysis of cells, cell aggregates, and physiology of the culture.

We would like to start this discission with a general statement that the in vitro methods/biotechnology provide evidence about the uniformity of the two different types of plants, spore plants and seed plants, which are characterized by different meristematic abilities in their natural habitats and in in vitro conditions. After selecting an established cell suspension, we provided evidence that, with specially designed in vitro experiments, it was possible to demonstrate numerous similarities between karyogamous plants (plants without secondary meristematic cell activity) and seed plants. Recently published papers investigating methods of somatic embryogenesis induction in tree ferns [12,41], based on the isolation of epidermal cells, laid the groundwork for the extension of in vitro experimentation with completely isolated cells from mother plant bodies, i.e., callus culture, and later for establishing that cell suspension culture. Due to its structural level, this type of culture system remains in contradiction to long-term gametophyte culture [4], which offers unlimited access to the experimental plant material very often used from the tree fern species. Gametophyte culture has resulted in spontaneous sporophyte production, and these subcultures might form the main source of the explants used, i.e., roots, fronds, leaves, and stolons from mature plants of non-tree-fern species [15].

In our experiments, root explants originating from the axenic culture of sporophytes produced a callus in the presence of 2,4-D and BAP. Callus proliferation on explants very seldom occurred. The established cell suspension was characterized by a relatively short period of 15 days, with irreversible darkening of the culture, which during the following 2 weeks led to necrosis of the sample in a particular conical flask. In additional experiments to determine the size parameter of the tissue sample, 3 g of tissue was indicated as the optimal amount. In the case of bigger samples of the tissue initiating the new subculture, the processes of culture necrosis occurred during a particular period of culture.

Nevertheless, some fern studies that do not include tree ferns have described roots and rhizomes as excellent sources of explants for callus induction in *Cyclosorus dentatus* [62], *Regnellidium diphyllum* [63] and *Drynaria quercifolia* [64] in the presence of auxins and cytokinins. In the case of *N. exaltata*, the culture was derived from an apical explant of stolon cultured in the presence of 2,4-D [15,16]. The callus that developed was used to establish cell suspensions of various shaped cells or small multicellular aggregates. Gametophyte-derived callus was used for the establishment of cell suspension cultures of the fern *P. coronarium* [17]. The results of our cell suspension establishment, with cell aggregates as the main cultured plant material, remained in contrast to the results described earlier [18,19] and recently by Riviera et al. [20]. Using liquid medium, a so-called “simple suspension culture” system of the cell was developed. The system consisted of a culture of various sized pieces of macerated leaf blade or gametophyte. Explants regenerated gametophytes that later produced sporophytes via apogamy [20]. In the case of our experiment with three-year-old *C. smithii* cell suspensions, no regeneration phenomenon was observed.

The established suspension presented the typical kinetics of growth reflecting the utilization of 2% of sucrose by 3.0 g of initial cell suspension tissue. Among the four sugars used (saccharose, maltose, glucose, and fructose) the highest value of tissue growth parameter was achieved with a fifteen-day-long subculture as the following decrease in fresh–dry–ash weight in relation to 1 > 10 > 100 for control culture. Studying the effect of various sources of organic carbon on the growth of *C. smithii* cell suspension, the following sugars were considered: sucrose (being the standard supplement of the medium of any type of in vitro culture), maltose (two molecules of D-glucose), and two monosaccharides, glucose and fructose, used separately [65]. The highest increase in all parameters studied (fresh, dry, and ash weight) was achieved in the presence of 3.0 g of inoculated tissue. An increase in the values of the three parameters (FW, DW, and ash) was observed independent of the type of studied sugar. Fructose as the monosaccharide appeared to have the strongest effect on all three parameters. The lowest effect was observed in the case of the culture maintained in the presence of maltose, which could indicate that its conversion in two molecules of D-glucose did not stimulate the growth and had a weaker effect on growth than glucose in all used concentrations. Double the number of glucose molecules did not stimulate the growth but appeared less favorable in comparison to fructose and itself in 20.0 g of glucose. Significant differences were recognized between fructose and glucose in comparing the growth of tissue. Studying the enzyme activity of the cell wall connected to the transfer of the studied sugars and their depletion in the cells will help to better understand the physiology of fern cells.

A main problem of tissue culture is the genetic stability of cultured plant explants and genetic evaluation of regenerants. In the case of the cell suspension carried out, cell suspension is a very dynamic system of cell culture with a change in all ingrediency during the culture period; even so, our results indicated that ethylene was regularly produced by the cultured cell suspension. The studied suspension being non-embryogenic and non-morphogenic confirmed the results published earlier on ETH production by a non-embryogenic type of cell suspension for a representative of the gymnosperm, *Picea glauca* [66]. In contrast to a solidified medium, in this condition the cell suspension culture includes three parameters that are used for the description of this phenomenon: (1) ETH production by the cells, (2) its solubility in the water medium, and (3) realization of ETH by the medium. The concentration of ethylene collected over the medium in a conical flask was correlated with the time of culture and the initial weight of *C. smithii* tissue. The measured amount of this organic was the resultant production by the cells and the solubility of ethylene in water (medium). The chemical data indicated 131 mg/L of the ethylene solubility in the water. Considering the problem of ETH production, the cell division activity, the culture rotation, the culture respiration, and the size of the aggregates should be taken under consideration. In our cultures 120 rpm by a rotary shaker guaranteed proper growth of the cell aggregates, which was confirmed by 3 years’ maintenance of established *C. smithii* culture. It seems that the enhancement of culture rotation results in the limitation of the size of aggregates and production of cell debris. Decreasing medium rotation will lower cell respiration, thereby reducing the dynamic of the cell growth.

In the presented paper the morphogenic phenomenon did not occur and only nuclear DNA content could be specified in the cells of aggregates. In a majority of flow cytometry studies, two fluorochromes are usually employed: propidium iodide and 4′,6-diamidino-2-phenylindole (DAPI). Without developing a method of protoplast isolation from cell suspension and osmotic lysis to achieve suspension of the nuclei, the direct use of fluorochrome acriflavine helped to compare the nuclear DNA content of cultured cells with this parameter of gametophytic and sporophytic cells [67]. The diploid value of cultured cells was validated. This result confirmed studies carried out on embryogenic cell suspensions of *Panicum maximum* and *Pennisetum purpureum* [68].

Using SEM tissue preparation, we were able to have a look inside the cells of the cultured aggregates, something never shown before for ferns. Initially, the cell suspension presented different regular cell divisions, giving evidence of elongated multicell structure formation; however, a well-established suspension was able to form only irregular aggregates consisting of different numbers of variously shaped and sized cells [16]. The physiological activities of the aggregated cells of the established suspension were expressed in their various morphological structure. The ½ MS medium assured the required level of minerals and 2% sucrose stimulated cell division and starch grain accumulation in the cell cytoplasm. SEM analysis indicated that starch accumulation should be connected with metabolic activity of the cell but not with its photosynthesis. We did not find evidence of plastid starch accumulation, which was confirmed by the lack of plastid differentiation, especially chloroplasts. The Sudan staining helped us to find some red globular structures indicating lipid bodies; however, numerous structures similar in size but not stained were also found in the cytoplasm. Tannins could alternatively be considered as forming globular bodies: tannosomes. Recently, chloroplasts were considered to be the site of tannin synthesis [69]. We proved this, but we were not able to define their organelle origin because of the lack of chloroplasts in cells of the established, yellow-colored suspension studied. Very rich flavonoid production was recognized as the yellow-colored medium, and cell aggregates themselves seemed very much related to the tannin condensation in vacuoles, based on the common machinery of flavan-3-ols as possible precursors of tannins. Using UV-light and laser excitation confirmed the tannins’ presence in the cells at various ages of the studied suspension observed in semi-thin and SEM specimens. We provided evidence of huge tannin bodies that were often observed in the cells from the specimens in our cytological analysis. The richness of the metabolite production of the cultured cells was confirmed by the yellow coloration of the medium from the beginning of establishment of the suspension to the present over at least 3 years, with each second-week subculture. Confocal analysis indicated the reach of the metabolic activity of the cytoplasm located close to the cell wall, which might be a visualization of the place of flavonoid production. When cells aged and lost mitotic activity, flavonoids were released via the cell wall into the liquid medium, giving it its yellow coloration. The intensity of the coloration was dependent on the quantity of soluble substances it contained, and it is likely that these were organic substances. Mass spectrometry showed that flavonol derivatives were responsible for the yellow coloration. These compounds are related to flavonoids and act as antioxidants, limiting the damage caused during oxidative stress [70]. The presence of these substances is easily recognized in senescent cells that have diminished capacity for cell division. The structure of the cell products was similar to that previously observed for the endosperm-derived callus of kiwi fruit [71] and wheat anther [72]. In the case of ferns, only *Pteris vittata* showed the extracellular matrix presence during callus proliferation of its prothallus [73].

Ferns are famous as traditional medicinal plants with attention paid to their flavonoids, radical scavenging, and antioxidant and anticancer activities. Flavonoids are secondary metabolites collected in all parts of the living plants [74], and in in vitro conditions are freely released from actively dividing cells of cell suspensions cultured in liquid medium, this being their response to stress [75] existing in the suspension. The intensive yellow coloration of the culture medium of the studied cell suspension indicated their rich secondary metabolism expressed by flavonoids production and was found in living samples of *Lygodium flexuosum* (climbing fern) and *Ampelopteris prolifera* (terrestrial fern in the tropics) [76], and Asplenioid ferns [77].

For the yellow coloration of the medium, the flavonoid content was used with quercetin (flavonol), presenting only a 301 nm extinction value [78]. The highest extinction value of quercetin was observed in the presence of fructose and was correlated with the increase in tissue mass of the inoculum. In the case of sucrose (the usual organic carbon source) non-significant differences were found; however, the highest value of extension was reached in the presence of 2 g of inoculated tissue. There are few references to flavonoid analysis in ferns. The rhizomes of the tropical small fern *Selliguea feei* [79] were the source of kaempferol-3-*O*-rutinoside, a bitter-tasting flavonol glycoside. Attention should be paid to cynarin, the natural product having a wide spectrum of biological activity including anti-HIV. It was recently established as a potent and highly selective class of HIV-1 integrase inhibitors [80]. The surfaces of the latter contain transparent structures that vary in shape and size. In the case of our unpublished data, in a long-term (6 month) liquid axenic multiplication culture of *C. delgadii*, yellow coloration of the medium was not observed. However, on the root surface numerous yellow deposits attached directly to the cells of the rhizodermis were observed. In many tests of its regeneration potential, a long-term cultured cell suspension implanted on the agar medium enriched by auxins and cytokinins never exhibited the yellow coloration of the solidified medium but showed dark (almost black) coloration of tissue, and the culture finally presented total necrosis.

Since ancient times ferns have been the object of human interest because of their consumer importance and rich secondary metabolism, later explored by the pharmaceutical industry [80] to investigate their underlying antibacterial properties. A majority of published papers have provided descriptions of living samples or herbaria, extraction being one objective of various methods of active substance isolation [81,82]. In the list of antibiotic activities of pteridophytes, among the 114 species studied, only *Cyathea crinite*, a tree fern, was considered with an analysis of its rhizome and sporophyll extracts against nine different bacteria [83]. Thirty-six species, including representatives of the *Cyathea* genus, did not inhibit any of the tested microorganisms, which was in contrast to the results we obtained for *C. smithii*. Our results demonstrated antibacterial activity against *Staphylococcus aureus*, *Escherichia coli*, and *Pseudomonas aeruginosa*. The results confirmed earlier studies finding the ethyl acetate extract of *Cyathea brunoniana* and *Pronephrium nudatum* fronds to be active against methicillin-resistant *Staphylococcus aureus* [82]. Moreover, the results for *C. smithii* were in accordance with [84], who showed the activity of *Cyathea microdonta* against Gram-positive bacteria. It was also noted that compounds isolated from the hexane extract of *C. nilgirensis* possessed antibacterial activity against *K. pneumoniae*, *E. coli*, *S. aureus*, and *B. subtilis* [85]. Nath et al. (2019) found that the ethyl acetate extract of *C. gigantea* possessed antibacterial activity, especially against *E. coli*, *P. aeruginosa*, and *S. aureus* [86]. These results were worse than those obtained for *C. smithii* cell suspension. The MIC values obtained in our study against *S. aureus*, *P. aeruginosa*, and *E. coli* were 3 to 51 times better than those obtained for *C. gigantea*. The antibacterial activity against *E. coli* and *S. aureus* of the *C. smithii* cell suspension was also much better than the hexane extract from mature fronds of *C. contaminans* [87]. Our results indicated the significant of antibacterial activity of *C. smithii*, among the other species of Cyatheaceae. Numerous representatives of fern exhibit a rich content of phenolic substances, which was shown in Malaysia flora, namely, *Cibotium barometz* and *Cyathea latebrosa* (both tree ferns) presenting antioxidant properties exhibiting strong DPPH radical scavenging activity [81].

## 5. Conclusions

This is the first published paper to study the long-term established cell suspension culture of ferns, including tree ferns. The culture was developed using callus proliferation derived from root explant. The paper provides evidence of the unique character of tree fern cell suspension on various levels (structural, cytological, and physiological). The presence of 2,4-D and BAP favored the establishment of differentiated types of cell suspension, which indicated the sporophytic value of the nuclear DNA content. We also provided evidence of the high metabolic activity of cultured cell suspensions (ETH and quercetin), which was illustrated by the microbiological tests against the various species of bacteria. Long-term culture provided the possibility of carrying on experiments involving the genetic manipulation of fern cells derived from cultured protoplasts. Furthermore, its modification enabled the regeneration of plants from established cell suspensions. Besides the numerous differences between ferns and spermatophytes, the systems of promulgating fern cells and their aggregates in liquid culture were found to be similar. Finally, we offered a new perspective on the potential benefits of increasing our knowledge of fern cells in the world of plants.

## Figures and Tables

**Figure 1 cells-11-01396-f001:**
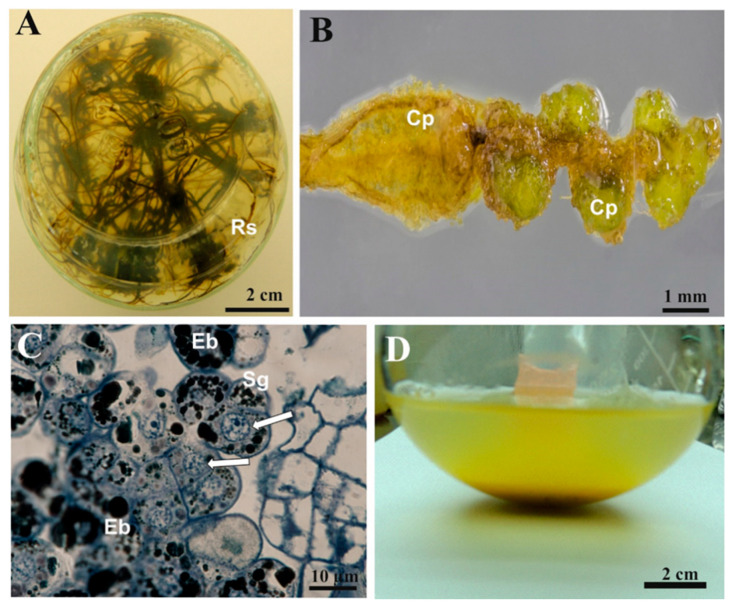
Callus induction and establishment of the cell suspension culture. (**A**) General view of root system of four-month-old axenic culture in jar. (**B**) Callus proliferation of root explant in the presence of 0.5 MS medium supplemented with 9.05 µM 2,4-D + 0.88 µM BAP and 2% sucrose after 6 weeks of culture. (**C**) Semi-thin section of callus showing small groups of cells containing nuclei in interphase, starch grains, and electron-dense bodies. Sections stained with toluidine blue (arrow). (**D**) General view of the conical flask containing the established cell suspension with yellow coloration of the medium. Cp—callus proliferation center; Eb—electron-dense body; N—nucleus; Rs—root system; Sg—starch grains.

**Figure 2 cells-11-01396-f002:**
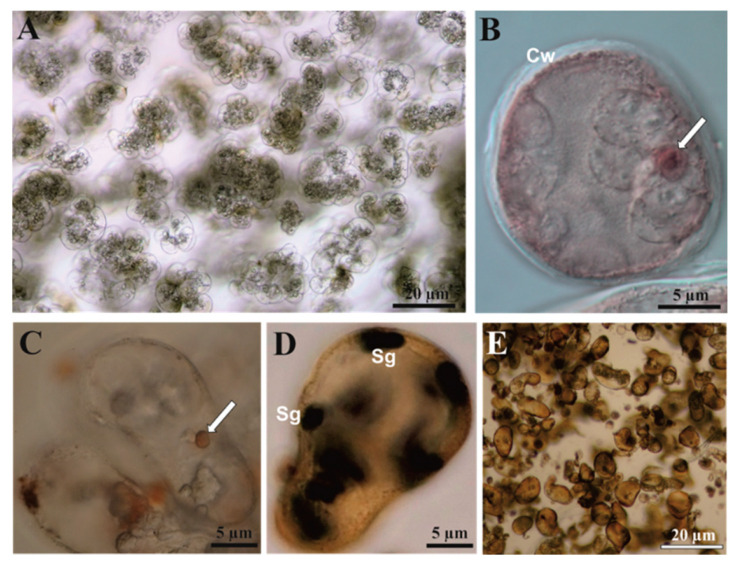
Cell suspension of *C. smithii* cultured in ½ MS medium supplemented with 9.05 µM 2,4-D + 0.88 µM BAP observed with Nomarski contrast. (**A**) General view of the cell aggregates. (**B**) Single cell with nucleus and distinct nucleoli stained with acetocarmine. (**C**) Lipid body (arrow) stained with Sudan III. (**D**) The cell showing numerous individual starch grains stained in black-blue coloration with in IKI. (**E**) Structure degradation of the ordered cell suspension by the subculture extension. Cw—cell wall, Sg—starch grains.

**Figure 3 cells-11-01396-f003:**
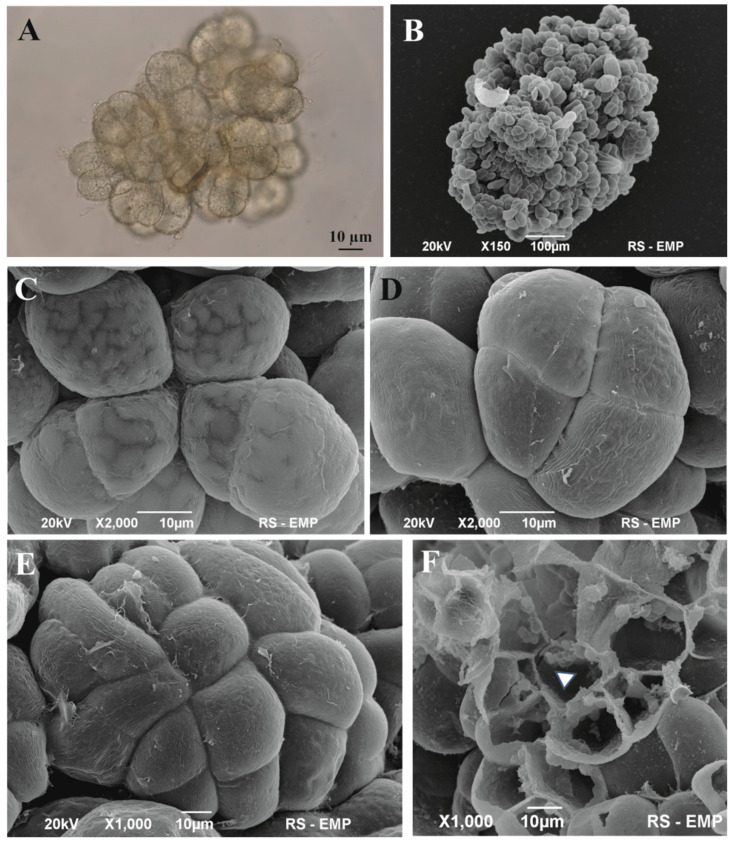
Analysis of established, differentiated types of cell suspension with evidence of some degree of cell organization in the structure of the *Cyathea smithii* cell suspension cultured in ½ MS medium supplemented with 2,4-D and BAP. (**A**) Cell aggregates observed with Nomarski contrast (squashed specimen). (**B**) SEM image of cell aggregate. (**C**) A thin cell wall facilitates the observation of large intracellular bodies. (**D**) Example of regular cell divisions forming a four-celled cluster. (**E**) Multicellular cluster with one centrally located cell regenerated on the surface of the cell aggregate. (**F**) Cross-section of a regular cluster regenerated from the cell aggregate with a centrally located triangular cell (white triangle) surrounded by eight cells of various locations, shapes, and sizes. (**A**) Bright-Field Light Microscopy; (**B**–**F**). SEM images.

**Figure 4 cells-11-01396-f004:**
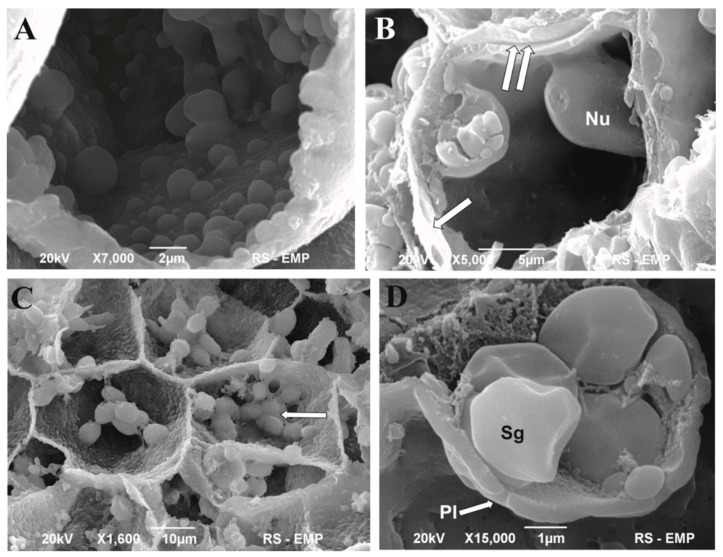
SEM image of the intracellular structure of cells of the *Cyathea smithii* cell aggregates. (**A**) An overview of the cell interior showing numerous organelles along the cell wall. (**B**) Cell wall (single arrow) and middle lamella (two arrows). (**C**) Globular cytoplasmic organelles (peroxisomes) associated with the cell wall or narrow bands of the cytoplasm. (**D**) Complex of starch grains surrounded by the cell wall with evident plasmodesmata. Pl—plasmodesmata; Nu—nucleus; Sg—starch grains.

**Figure 5 cells-11-01396-f005:**
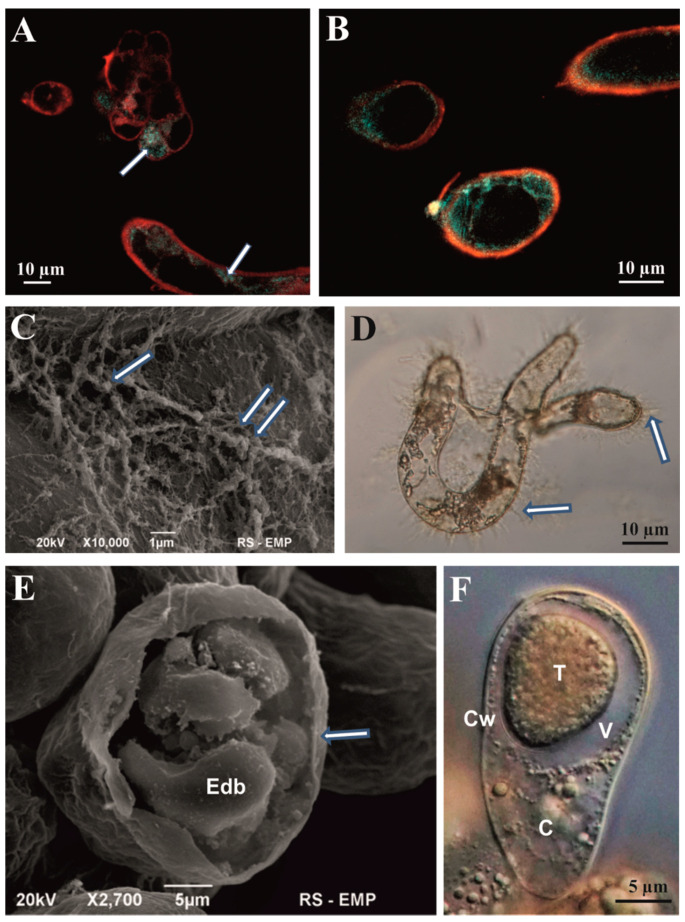
Structural evidence of the cell suspension metabolic activity. (**A**) Autofluorescence of small, actively dividing cells of the cell aggregate and of long differentiated cells; arrows show the condensation of tannins. (**B**) Autofluorescence of single cells. (**C**) Cell wall coated by a “net” or reticulum of granular (single arrow) and filamentous (double arrow) metabolites released by the cell. (**D**) Active production of metabolites by non-mitotically active cells (the transparent filaments) and their release outside the cell. (**E**) Opened marginal cell of aggregate in “cross-section” showing electron-dense bodies. (**F**) Large dense body located in a number of dense bodies. Cw—Cell wall; V—Vacuole; C—Cytoplasm; T—Body of tannins. (**A**,**B**)—confocal microscopy; (**C**,**E**)—scanning electron microscopy; (**D**,**F**)—light microscopy with Nomarski contrast (squashed specimen).

**Figure 6 cells-11-01396-f006:**
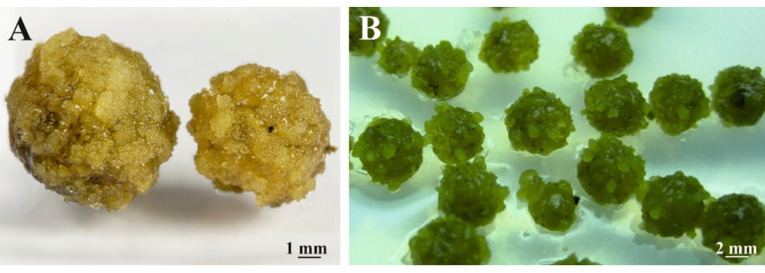
Rich cell proliferation of the cell suspension aggregates packed in the alginate capsules, after 4 weeks of the culture in MS medium supplemented with 2,4-D + BAP (**A**) and an application of encapsulated cell suspension for plant growth hormone tests (**B**). Insular tissue proliferation noted but lacking any evidence of the organ differentiation (5×).

**Figure 7 cells-11-01396-f007:**
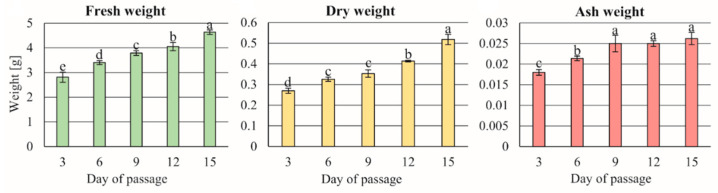
The kinetic of the growth of the cell suspension in a standard culture of liquid ½ MS medium supplemented with 9.05 µM 2,4-D + 0.88 µM BAP and 20.0 g/L sucrose.

**Figure 8 cells-11-01396-f008:**
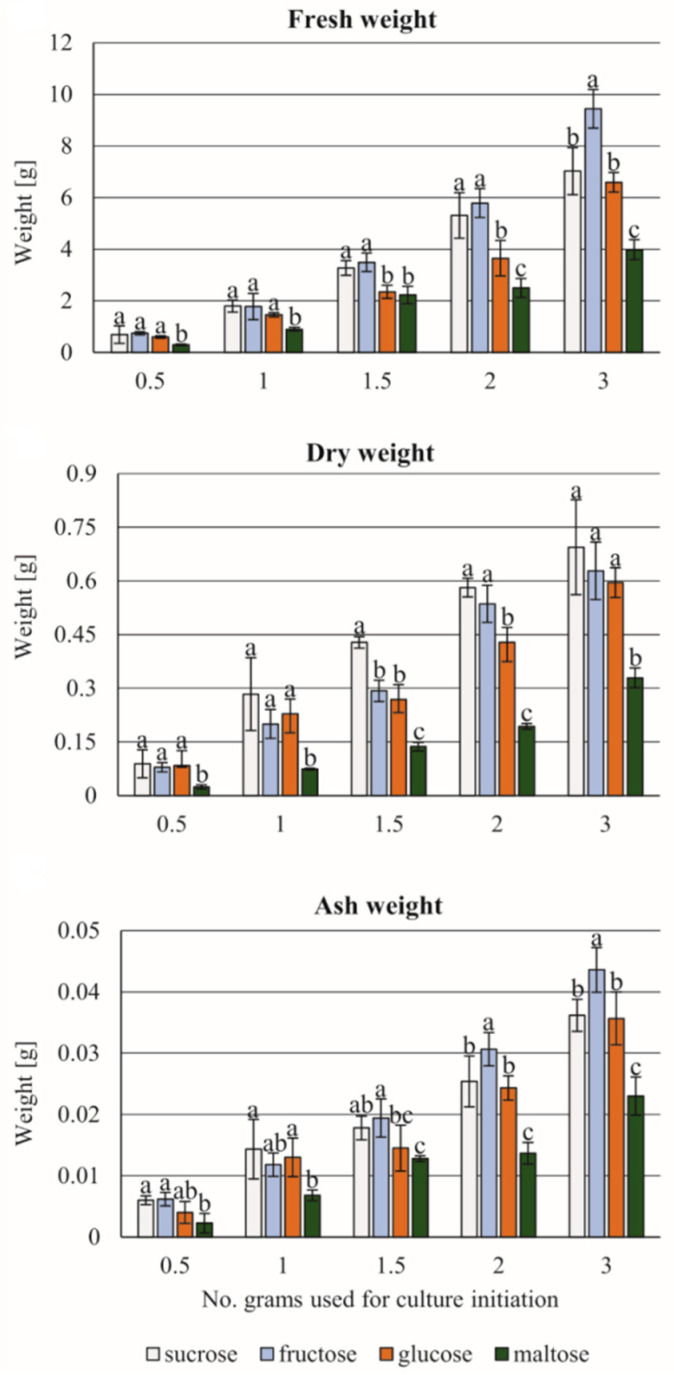
The dynamic growth of the cell suspension according to the initial weight of tissue and the saccharides supplementing the ½ MS medium with 9.05 µM 2,4-D + 0.88 µM BAP. Note: fructose has a significant effect on the dynamic of tissue growth. Means ± standard deviations marked by different letters are significantly different (*p* ≤ 0.05).

**Figure 9 cells-11-01396-f009:**
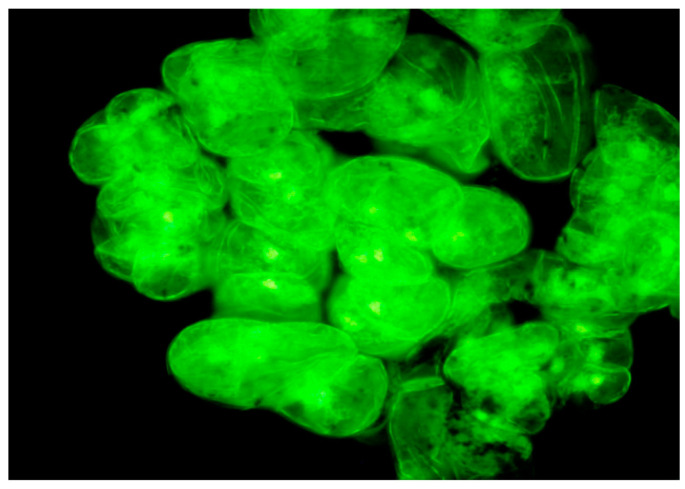
The image of nuclear DNA staining of the cell suspension with the help of DNA-specific fluorochrome acriflavine with acetic hydrolysis (400×).

**Figure 10 cells-11-01396-f010:**
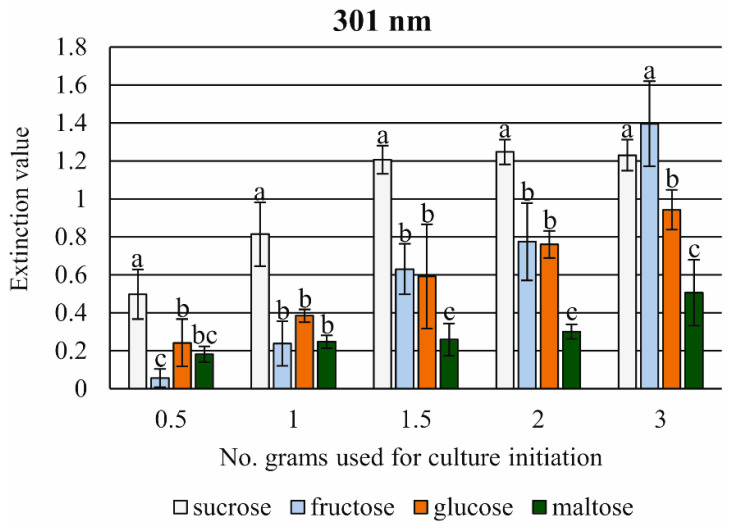
The effect of 20.0 g of saccharides and initial weight of cell suspension tissue on the extinction value of quercetin measured at 301 nm wavelength in the post-culture medium after a two-week-long subculture. Means ± standard deviations marked by different letters are significantly different (*p* ≤ 0.05).

**Table 1 cells-11-01396-t001:** Effectivity of ETH production in relation to the FW, DW ASH of initial tissue according to the length of passage time for 80 mL of ½ MS medium supplemented with 9.05 µM 2,4-D + 0.88 µM BAP.

Sample No.	Time of Cultivation (Days)	Amount of ETH (ppm)	Amount of ETH per mg FW	Amount of ETH per mg DW	Amount ETH per mg of ASH
1.	3	5.64 ± 0.00	1.77 ± 0.05 ^a^	17.80 ± 0.40 ^a^	0.0035 ± 0.0002 ^a^
2.	6	6.67 ± 0.00 ^a^	1.34 ± 0.04 ^b^	17.88 ± 3.12 ^a,b^	0.0042 ± 0.0001 ^b^
3.	9	5.34 ± 0.00	1.63 ± 0.04 ^a,b,c^	16.02 ± 4.39 ^a,b,c^	0.0046 ± 0.0003 ^c^
4.	12	5.56 ± 0.00	0.53 ± 0.02 ^c^	6.02 ± 0.29 ^a^	0.0096 ± 0.0007
5.	15	6.68 ± 0.00 ^a^	1.51 ± 0.48 ^a,b,c^	23.30 ± 6.85 ^a,b,c^	0.0044 ± 0.0016 ^a,b,c^

Means ± standard deviations marked by different letters are significantly different (*p* ≤ 0.05). The same letters indicate the lack statistical differences between results. ETH—ethylene; FW—fresh weight; DW—dry weight.

**Table 2 cells-11-01396-t002:** Effectivity of ETH production in relation to the initial weight of used of cell suspension for 40 mL of ½ MS medium supplemented with 9.05 µM 2,4-D + 0.88 µM BAP.

Sample No.	Initial Weight (g)	Amount of ETH (ppm)	Amount of ETH per mg FW	Amount of ETH per mg DW	Amount ETH per mg of ASH
1.	0.5	5.25 ± 0.00	7.05 ± 0.25	58.00 ± 3.05	0.0006 ± 0.0001
2.	1.0	5.46 ± 0.00	3.25 ± 0.04	26.05 ± 0.68	0.0021 ± 0.0001
3.	1.5	5.51 ± 0.00	2.20 ± 0.10	16.50 ± 0.95 ^a^	0.0031 ± 0.0001
4.	2.0	5.53 ± 0.00	1.20 ± 0.04	12.50 ± 0.31 ^a^	0.0045 ± 0.0002
5.	3.0	4.55 ± 0.00	0.61 ± 0.48	10.25 ± 4.68 ^a^	0.0083 ± 0.0004

Means ± standard deviations marked by different letters are significantly different (*p* ≤ 0.05). The same letters indicate the lack statistical differences between results. ETH—ethylene; FW—fresh weight; DW—dry weight.

**Table 3 cells-11-01396-t003:** Zones of bacterial growth inhibition as antibacterial *C. smithii* extract samples.

Sample	*Staphylo* *coccus aureus*	*S. epidermidis*	*Escherichia coli*	*Pseudo* *monas aeruginosa*	*S. mutans*	*Propioni* *bacterium acnes* *PCM 2400*	*P. acnes* *PCM* *2334*	*S. sanguinis PCM 2335*
	Aerobic Gram-Positive		Aerobic Gram-Negative		Microaerobic Gram-Positive			
**Standards**								
quercetin	0	0	0	0	0	0	4	0
luteolin	0	0	0	0	0	0	0	0
gallic acid	15	36	12	10	16	20	22	18
*p*-coumaric acid	0	0	8	6	0	0	0	0
caffeic acid	0	28	4	4	0	0	0	0
vanillic acid	0	0	4	4	0	0	0	0
**Samples of *Cyathea smithii***								
In vitro ***	0	0	0	0	0	0	0	0
Green house **	0	0	0	0	0	6	6	0
Cell suspension ***	15	10	40	36	19	18	16	22

Sensitivity of: *—in vitro-derived regenerant, **—fragment of the leaf blade from a several years old plant, ***—cell suspension maintained in the presence of ½ MS medium supplemented with 9.05 µM 2,4-D + 0.88 µM BAP.

**Table 4 cells-11-01396-t004:** The minimum inhibitory concentration (MIC, μg/mL) and MBC/MIC ratio as an antibacterial activity of the studied *C. smithii* extracts.

Sample	*S. aureus ATCC* 25923	MBC/MIC	*S. epidermidis ATCC 12228*	MBC/MIC	*E. coli ATCC* 25992	MBC/MIC	*P. aeruginosa ATCC* 27853	MBC/MIC	*S. mutans PCM 2502*	MBC/MIC	*P. acnes PCM 2400*	MBC/MIC	*P. acnes PCM 2334*	MBC/MIC	*S. sanguinis PCM 2335*	MBC/MIC
**Cell ***	62.5	8	125	16	7.8	16	62.5	8	62.5	4	62.5	4	62.5	4	15.6	2
**Gallic acid**	125	16	31.25	8	500	8	1000	-	125	8	62.5	8	62.5	8	62.5	8

* Cell suspension maintained in the presence of ½ MS medium supplemented with 2,4-D + BAP.

## Data Availability

The data sets generated for this study are available on request from the corresponding author.

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
