# Peer review of "Biotechnology of the Tree Fern Cyathea smithii (J.D. Hooker; Soft Tree Fern, Katote) II Cell Suspension Culture: Focusing on Structure and Physiology in the Presence of 2,4-D and BAP"

_cells, 2022, doi:10.3390/cells11091396_

Round 1

Reviewer 1 Report

Reviewer #01 (Report 01)

This research study the RybczyÅ„ski et al., on the article research (cells-1655212) entitled “Biotechnology of the Tree Fern Cyathea smithii (J.D. 2 Hooker1854; Soft tree fern, Katote). II. Cell Suspension Culture: 3 Focusing on Structure and Physiology in the Presence of 2,4-D 4 and BAP”. The present work is valuable and informative for the demonstrated effects of new report of establish cell suspension based a MS medium and 2,4-D and BAP with tree ferns species. The research demonstrated the potential production, yield, and benefits to the environment and agriculture with importance applied to soil, plant and humans, and your beneficious a bactericidal and bacteriostatic production based in this new in vitro culture and others points. Introduction its ok, but M&M, results and discussion still need to improve some points. The material and methods was good described. However, statistical description its many corrections is necessary (include a topic). Also, senteses and synthases is requerid. The data its relevant and the manuscript, results and discussion, table and figures showed its good presentation. The paper is concise and relevance to scopus of the scopus journal. However, major points and English grammar its necessary to expert native to improve a reader to Cells journal.

To make it easier for the authors, I made notes, questions and comments described in the pdf attached with major and minor points.

A second point, observe and standardize the terms throughout the text.

The third point, what is the purpose of using 2,4-D and BAP. Perhaps this point has not been explained clearly, although I know and understand the relevance, perhaps a better explanation along with the discussion, is relevant to reach the largest number of Cells readers.

Best Regards

Reviewer 2 Report

The reviewed article describes cell suspension culture of tree fern Cyathea smithii obtained for the first time by the authors. The article is supplied with beautiful pictures illustrating some details of cells structure, ploidy, presence of secondary metabolites and other characteristics obtained with the help of scanning electron, confocal microscopy, histochemistry and other methods. The article is likely to be interesting for readers. Still, from my point of view, the text should be thoroughly revised before it may be published. Some descriptions are not clear enough. I noticed a lot of unclear sentences and mistakes in English. All of them are marked in the pdf file I am going to attach to my report. I also proposed how to modify some of them. But, since English is not my first language, I strongly advise authors to seek the help of professional English language specialist. Here I provide only part of my remarks. All the rest should be found in the attached file and carefully addressed.

  1. Lines 28-29. “Microbiological studies suggested that bactericidal character possessed cell suspension against microaerobic Gram-positive bacteria” – there is a mistake in succession of the words. I suggest “cell suspension possessed bactericidal character against microaerobic..."
  2. Line 37-38. “aerial layering are stick to their trunk, what helps to form roots” – “what” should be substituted with “which” or “that” throughout the text. Next, authors should explain what is “aerial layering”. This structure is sometimes called “aerial roots” and if this is so, “helps to form roots” is out of place here (aereal roots help to form roots).
  3. Line 40-41. “The number of the papers concerning plant cell manipulations is limited only to few species” – it should be specified, which plant species are meant. There are a lot of papers concerning cell culture of numerous plant species.
  4. Lines 81-82. “plant material is completely independent of plant growth hormone conditions.” – I cannot agree with “completely”. There are a lot of reports showing importance of genetically controlled capacity of plants to metabolize and absorb hormones for the outcome of the cultivation in vitro.
  5. Study of ethylene production by cell suspension should be mentioned in the aim of the research.
  6. It is frequently unclear and should be checked through the text, which mass is meant (initial, or the mass got at the end of cultivation).
  7. Sections 3.5. and 3.5. There are no references to illustrations in these sections, but it looks as if table 4 is described in 3.5 and table 3 – in 3.5. The order of description should be rectified.
  8. Lines 499-500. “This fact indicated that cells with lower initial weight of cells produced low amount of ETH but those with greater initial weight of cells produced lower amount of ETH.” - First part of the sentence contradicts with the second. It is unclear in which case ET production was really lower (with lower or greater initial weight).
  9. Lines 505-507. “It should be noted that a final amount of plant material expressed in FW and DW do not reach about 10 g and about 1 g, respectively, like it was in period of the culture (Fig. 9).” – one more unclear sentence. It is unclear and should be specified which period of culture is meant.
  10. Table 4. Something is seemingly wrong here: 0.0096 ±01. The SE is greater than the mean value.

All my other remarks should be found in the comments to the file. I hope they will be visible.

Round 2

Reviewer 1 Report

Dear Author, all considerations have been answered. I think manuscript can be accepted. Best Regards

Reviewer 2 Report

Authors carefully addressed all my remarks. I have no obections against its publication in present form.